# Comparison of Four Systems to Test the Tolerance of ‘Fortune’ Mandarin Tissue Cultured Plants to *Alternaria alternata*

**DOI:** 10.3390/plants10071321

**Published:** 2021-06-28

**Authors:** Margarita Pérez-Jiménez, Olaya Pérez-Tornero

**Affiliations:** Equipo de Mejora Genética de Cítricos, Instituto Murciano de Investigación y Desarrollo Agrario y Alimentario (IMIDA), 30150 Murcia, Spain; margarita.perez3@carm.es

**Keywords:** brown spot, ACT, fungus culture filtrate, mycotoxin

## Abstract

Alternaria brown spot is a severe disease that affects leaves and fruits on susceptible mandarin and mandarin-like cultivars, and is produced by *Alternaria alternata*. Consequently, there is an urge to obtain new cultivars resistant to *A. alternata*, and mutation breeding together with tissue culture can help shorten the process. However, a protocol for the in vitro selection of resistant citrus genotypes is lacking. In this study, four methods to evaluate the sensitivity to *Alternaria* of mandarin ‘Fortune’ explants in in vitro culture were tested. The four tested systems consisted of: (1) the addition of the mycotoxin, produced by *A. alternata* in ‘Fortune’, to the propagation culture media, (2) the addition of the *A. alternata* culture filtrate to the propagation culture media, (3) the application of the mycotoxin to the intact shoot leaves, and (4) the application of the mycotoxin to the previously excised and wounded leaves. After analyzing the results, only the addition of the *A. alternata* culture filtrate to the culture media and the application of the mycotoxin to the wounded leaves produced symptoms of infection. However, the addition of the fungus culture filtrate to the culture media produced results, which might indicate that, in addition to the mycotoxin, many other unknown elements that can affect the plant growth and behavior could be found in the fungus culture filtrate. Therefore, the application of the toxin to the excised and wounded leaves seems to be the most reliable method to analyze sensitivity to *Alternaria* of ‘Fortune’ explants cultured in vitro.

## 1. Introduction

*Alternaria alternata* is a fungus that can produce severe damage in economically important plants, including cereal crops, vegetables, and fruits. *A. alternata* is able to infect plants producing host-selective toxins during the germination of spores on plant surfaces. Taking these toxins into consideration, seven pathotypes can be distinguished [1]. *A. alternata* pv. citri is the specific pathotype for *Citrus reticulata* and its hybrids, and produces the host-specific Alternaria citri toxin (ACT), which provokes, in sensible mandarin trees, the emergence of Alternaria brown spot (ABS), a severe disease whose main features are lesions on leaves and immature fruits of the infected plants that can result in early fruit abscission [2]. This reduces the quality and the commercial value of fruits in the market, leading to huge economic losses globally every year [3].

‘Fortune’ mandarin (‘Fina’ clementine × ‘Dancy’ mandarin) is a *Citrus reticulata* hybrid. Although this cultivar is widely harvested in Spain due to the excellent characteristics of its fruits, it is sensible to ABS. Thus, obtaining new ABS-resistant cultivars with the ‘Fortune’ organoleptic and farming qualities would be desirable for the mandarin market. In this sense, mandarin breeding programs are trying to include these features in their new selections. However, citrus breeding programs using conventional methods involve several difficulties associated with long juvenile periods, high heterozygosity, polygenic traits, and complicated genetic systems [4]. To overcome these difficulties, breeders have developed mutation breeding techniques [5], which could lead to a new cultivar with the desirable qualities of ‘Fortune’, but excluding the sensitivity to ABS.

If mutation breeding is increasingly considered to be a powerful alternative for the generation of genetic variations for plant breeding in citrus, tissue culture offers the possibility of managing large populations in a limited space while allowing a more rigorous control of the environmental conditions [6] and reducing the time spent until the selection of a genotype can be performed. To make a selection in ex vitro plants, waiting until they are grown to evaluate them is imperative. However, tissue culture offers the possibility of making an early selection of plant material, which saves space, time, and human resources [7]. Hitherto, a tissue culture protocol for mutant selection has been developed in potato and apple for two different species of the genus *Alternaria* [8,9] and in sugarcane for *Fusarium sacchari* [10]. However, there is no in vitro selection protocol for genotypes that are resistant to *Alternaria alternata* in citrus, and this is key to evaluate mutants developed in vitro.

In this study, four systems aimed to test the tolerance of ‘Fortune’ mandarin tissue cultured plants, which are very sensitive to ABS, to *A. alternata* were compared. Sensitivity to an *A. alternata* culture filtrate, as well as the ACT, was tested.

## 2. Results

### 2.1. ACT or Culture Filtrate Addition to the Culture Media

After four weeks of culture, no effect of the ACT in the culture media was observed in the ‘Fortune’ shoots. Actually, no significant differences were found either in the proliferation rate, length, productivity, or leaf damage with any treatment (data not shown).

When the *A. alternata* culture filtrate was added to the culture media, ‘Fortune’ explants were significantly (*p* < 0.0001) affected by the filtrate volume in the culture media in all of the studied parameters. Proliferation rate decreased from 2.06 in the control to 1.25 when the fungus culture filtrate was 50% of the volume in the culture media, and significant differences were observed between the three concentrations (Figure 1). Likewise, productivity decreased with the presence of the *A. alternata* culture filtrate in the culture media (Figure 1), although significant differences were not observed between the control and 25% of the culture filtrate. On the contrary, the addition of the culture filtrate to the culture media boosted the shoots average length from 11.29 mm in the control to 13.55 mm with 50% of the culture filtrate, and significant differences were not observed for 25% and 50% (Figure 1).

Finally, the culture of ‘Fortune’ shoots in vitro by adding the *A. alternata* culture filtrate was significantly affected (*p* < 0.0001) by the filtrate volume in the culture medium, producing damage in around three leaves by explant in shoots cultured with a 25% of the filtrate and around eight leaves by explant in the shoots in a 50% of the fungus filtrate (Figure 1).

### 2.2. ACT Application to Shoot Leaves

Regarding the ACT application to in vivo leaves, results were similar to those obtained with the ACT addition to the culture media, so there were no effects. Thus, after four weeks of culture, no significant differences were found between leaves in shoots grown in the control vessels (0 mL L^−1^ of ACT) and the rest of the treatments (Figure 2; data not shown).

Direct application of ACT in the ‘Fortune’ wounded excised leaves has a significant effect (*p* < 0.0001) over the percentage of damaged leaves. The mycotoxin caused a significant damage when the concentration of the ACT was 75 mg L^−1^ (Figure 3 and Figure 4). At this concentration, almost 80% of the leaves per shoot were damaged. Although some leaves were damaged at 25 and 50 mg L^−1^ of ACT, no significant differences were found between these two treatments and the control.

## 3. Discussion

ABS is a severe disease on susceptible mandarin and mandarin-like cultivars [11]. Thus, the obtention of new genotypes that preserve good organoleptic qualities, along with a medium-to-high tolerance to the infection by *A. alternata*, will entail a huge advance in the mandarin culture. On the other hand, citrus breeding involves a costly and long technical effort that could be shortened (and made more cost-effective) with mutation breeding and tissue culture, a technique that has proved to be very efficient in the early selection of citrus genotypes [12]. In this study, we have tested four different methods to detect *A. alternata* susceptibility in vitro in order to set a fast and efficient protocol for plant selection.

The methods studied in this work can be divided into two categories: (1) modification of the culture media by the addition of *A. alternata* components (fungus culture filtrate or extracted ACT), and (2) the application of ACT to leaves excised from shoot culture. When ACT was added to the culture media, no differences between the control and the treatments were detected, so no toxicity emerged from the application of ACT at the used concentrations to the shoots in this experiment. Thus far, there is no information about the addition of ACT to media in citrus shoots in previous reports, and it is also scarce in other species. Only Chakraborty et al. (2020) [13] reported induced-tolerance in shoots of *Withania somnifera* to *A. alternata* by exposing calli to a media containing mycotoxin isolated from an *A. alternata* culture. The differences between this study and the current trial can be found in the concentration of the toxin and the sensitivity of certain tissues and species to the toxin. Thus, a higher sensitivity to the *A. alternata* toxin has been found in calli than in shoots, so a higher concentration is needed in shoots to replicate the effects produced in calli [14].

On the contrary, when the culture filtrate of *Alternaria* was added to the culture media, a toxic response was observed in the shoots (Figure 1). As it was revealed in this experiment, the capacity of in vitro shoot multiplication in citrus, as well as the number of damaged leaves, are clearly affected by any biotic [15] or abiotic stress [7]. However, the shoots length was boosted by the *A. alternata* culture filtrate in the culture media. This last result would suggest that the filtrate could contain many other unknown compounds, beyond ACT, that might be also playing positive or negative roles in the in vitro culture of ‘Fortune’, separately and/or synergistically with ACT. The fungal culture filtrate has been used by few authors for inducing resistance to some species of the *Alternaria* genus through tissue culture techniques [9,14,16,17]. The components of this filtrate were not analyzed in any of these studies; hence, the exact content of the culture media, and even the presence of mycotoxin and its concentration, were actually unknown. Nevertheless, pathogen culture filtrates are part of the selection protocol most commonly used for in vitro production and selection of disease-tolerant plants in many crops [18]. Moreover, although the exposition to the products of the fungus activity in the filtrate applicated via culture media has proved to be efficient for the induction of resistance to many pathogens, due to the big amount of components of the culture. Fungus culture would not probably be the best method to evaluate *A. alternata* resistance in vitro, since the infection by *A. alternata* comes into play during the germination of spores on plant surfaces and not through the absorption of the pathogen [19]. The simulation of an infection should be as similar as possible to the real process not to introduce new factors that can alter our results.

In contrast, the application of the toxin to the leaves would be more similar to the traditional and efficient ex vitro test and also to naturally-occurring infections. This application was tested in intact leaves in the shoot and in wounded leaves excised from the shoot. In the case of intact leaves, no damage was observed, and the opposite was noticed in the wounded leaves, since above 75% of the leaves presented damage with an application of a preparation containing 75 mg L^−1^ of ACT. Similar results were obtained by [8] with chemically-synthesized AM-toxin I of *A. alternata* in in vitro leaves obtained from apple mutant shoots, and the same method was used by [20] in *Vitis vinifera* to evaluate the capacity of vitis cultivars to resist the *Erysiphe necator* infection. This protocol had not been proven in vitro in citrus plants to test them against ACT before; however, it has been extendedly used ex vitro thanks to the studies made by [2] in *A. alternata* pv. *citri*.

## 4. Materials and Methods

### 4.1. In Vitro Material

Plant material was obtained from shoot cultures of ‘Fortune’ mandarin established in vitro and subcultured monthly on proliferation media composed of MS salts and vitamins [21] supplemented with 2 mg L^−1^ of 6-benzylaminopurine, 0.1 mg L^−1^ of indolebutyric acid, 0.6 mg L^−1^ of gibberellic acid, 30 g L^−1^ of sucrose and 7 g L^−1^ of agar (Hispanlab), as described by [22]. After adding plant growth regulators and adjusting the medium pH to 5.7 with 1 N NaOH, 100 mL of medium were dispensed in each of the 500-mL jars and sterilized in an autoclave at 121 °C for 21 min. Cultures were grown at 25 ± 1 °C with white light (5000 lux) and a 16-h photoperiod.

### 4.2. A. alternata Isolates and ACT Purification

Isolates of *A. alternata* pv. *citri* were obtained from fruits affected by ABS in a commercial plantation of ‘Fortune’ mandarin as described by Nemsa et al. (2012) [23]. Likewise, the fungus culture, pathogenicity studies, and ACT purification were conducted as described by del Río et al. (2018) [24]. Briefly, cultures in Petri dishes of the selected *A. alternata* isolates (15 day-old) were used to extract ACT. Cultures were dried in an oven at 60 °C and grounded before ACT extraction with acetonitrile (1 g 10 mL^−1^) as a solvent, while stirring for 2 h.

### 4.3. A. alternata Culture Filtrate Preparation

The *A. alternata* culture filtrate preparation protocol was adapted from Kumar et al. (2008) (14). A virulent *A. alternata* pv. *citri* strain was inoculated initially in Petri plates containing PDA (potato dextrose agar) medium at 27 °C. After 10 days of culture, a mycelium portion of 5 × 5 mm was inoculated in an Erlenmeyer flask with liquid MS medium and 3% of sucrose. Cultures were maintained in agitation (125 rpm) and darkness at 25 ± 2 °C for 50 days. After that period, the mycelium was homogenized with a glass rod and filtered through a Whatman Grade 1 paper filter with a vacuum flask to obtain an *A. alternata* filtrate deprived from spores and mycelium. The fungus culture filtrate was centrifugated at 5000 rpm for 20 min and filtrated once again using a Whatman Grade 1 paper filter. The pH of the resulting filtrate was adjusted to 5.7, and the solution was stored at −20 °C in brown bottles.

### 4.4. ACT Addition to the Culture Media

Shoots of ‘Fortune’ from proliferation were cultured in glass tubes (150 × 20 mm) containing 15 mL of proliferation medium with different concentrations of ACT (0, 1, and 2 mg L^−1^), previously filter-sterilized using a nylon filter with a membrane pore size of 0.45 µm, plus an additional control with water to know the possible effects of acetonitrile. The experiment consisted of 20 glass tubes per treatment. Medium sterilization was performed and culture conditions were as described previously.

After 4 weeks of treatment, the number of shoots (longer than 5 mm) per explant, their average length, and the number of damaged leaves were recorded. From these data, proliferation rate (no. shoots/explant) and productivity (shoot average length × proliferation rate) were calculated. The number of damaged leaves included not only the leaves that appeared chlorotic or with necrotic spots, but also the fallen leaves that remained on the culture media.

### 4.5. Culture-Filtrate Addition to Leaves Excised from Shoot Culture

Filtrate (1 mL) was previously cultured in PDA Petri dishes for 7 days at 27 °C to ensure sterility. Media was prepared as a proliferation media, but adding the corresponding percentage of filtrate to reach a final volume of 100 mL per vessel, after sterilization in the autoclave. Thus, 3 treatments were studied: 0%, 25%, and 50% of filtrate volume in the culture media. Medium sterilization and culture conditions were performed as described previously. The experiment consisted of 6 glass vessels per treatment with 6 shoots each. Evaluation criteria were similar to those of the previous experiment.

### 4.6. ACT Application to Shoot Leaves

ACT was applied to in vivo leaves or wounded leaves in different experiments. Firstly, filter-sterilized ACT was diluted in acetonitrile to the concentrations of 0, 10, 100, and 500 mg L^−1^, and an additional control with water was prepared in order to check the possible effects of acetonitrile. ACT was applied with a brush, previously sterilized in the autoclave, to the first and third fully-expanded in vivo leaves of 20 in vitro shoots of ‘Fortune’ per ACT treatment. The explants were cultured in vitro for 4 weeks after the application of the treatment, and after that time the number of damaged leaves was recorded.

Additionally, using the previously-described protocol, filter-sterilized ACT at the concentrations of 0, 12.5, 25, 50, and 75 mg L^−1^ was applied with a brush to the first 4 fully-expanded leaves of ‘Fortune’ explants. Leaves were first excised from the shoots and placed in sterile Petri plates with a humid filter paper. Subsequently, leaves were wounded in the abaxial side to improve mycotoxin penetration, and ACT was applied with a sterilized brush. Four repeats (plates) and 10 leaves per plate were used per treatment. The results were evaluated a week after the application of the treatment, and the percentage of damaged leaves was recorded.

### 4.7. Statistical Analysis and Data Presentation

Data were first tested for homogeneity of variance and normality of distribution. Significance was determined by analysis of variance (ANOVA), and the significance (*p* < 0.05) of any differences between mean values was tested by Duncan’s new multiple range test, using Statgraphics Centurion^®^ XVI (StatPoint Technologies Inc., The Plains, VA, USA).

## 5. Conclusions

In the present study, four methods to evaluate the susceptibility of mandarin cultivar ‘Fortune’ explants to *A. alternata* pv. *citri* have been tested. After analyzing the results and the previous studies, the adaptation to the in vitro environment of the traditional method of excising and wounding leaves seems to be the most reliable approach. According to our results, a higher concentration of ACT could be necessary to obtain symptoms in the in vitro cultured shoots, and the fungus culture filtrate, although widely used, may contain many other components rather than ACT that might introduce noise in the results. In conclusion, the wounding in the leaves seems to be basic in order to infect the leaf surface. Thus, the selection of mandarin genotypes resistant to ABS can be better performed in vitro trough the excision of leaves of the studied shoots and after producing small wounds in the leaf surface. This first report about an in vitro protocol for the selection of mandarin genotypes resistant to *A. alternata* will provide a basic tool to produce and select mutants resistant to *A. alternata* in citrus.

## Figures and Tables

**Figure 1 plants-10-01321-f001:**
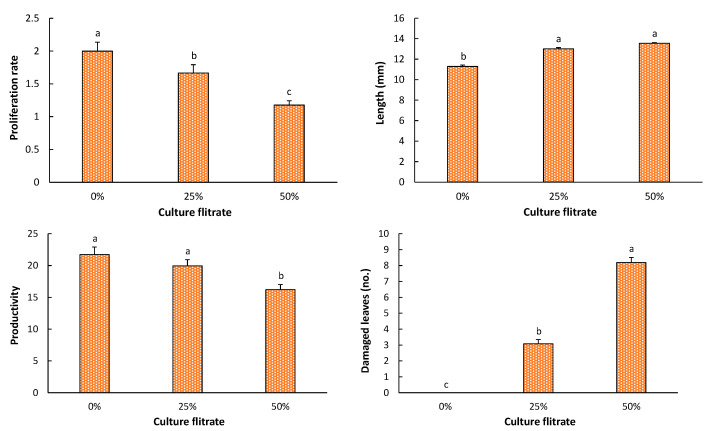
Proliferation rate, elongation, productivity, and number of damaged leaves in shoots of the mandarin cv. ‘Fortune’ exposed to different percentages of culture filtrate of *Alternaria alternata*. Data represent average ± SD values. Bars with different lower-case letters indicate a significant difference according to the LSD test (*p* ≤ 0.05).

**Figure 2 plants-10-01321-f002:**
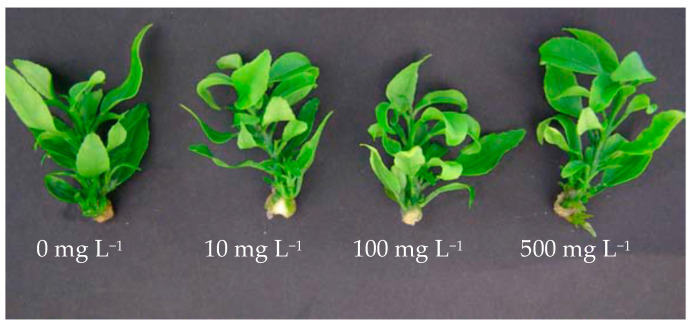
Shoots of the mandarin ‘Fortune’ grown at different concentrations of *Alternaria citri* toxin.

**Figure 3 plants-10-01321-f003:**
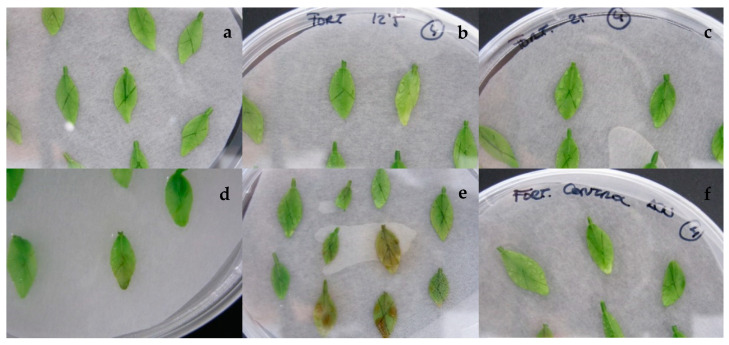
Excised and wounded leaves of the mandarin ‘Fortune’ exposed to 0 (**a**), 12.5 (**b**), 25 (**c**), 50 (**d**), and 75 (**e**) mg L^−1^ of *Alternaria citri* toxin and acetonitrile (**f**).

**Figure 4 plants-10-01321-f004:**
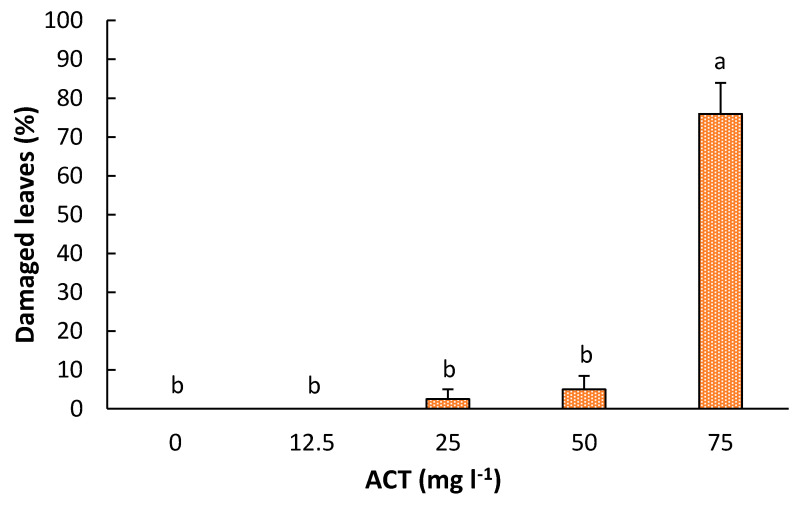
Percentage of damaged leaves exposed to a medium containing different concentrations of *Alternaria citri* toxin (ACT). Data represent average ± SD values. Bars with different lower-case letters indicate a significant difference by LSD test (*p* ≤ 0.05).

## Data Availability

Not applicable.

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
