# Peer review of "Comparison of Four Systems to Test the Tolerance of ‘Fortune’ Mandarin Tissue Cultured Plants to Alternaria alternata"

_plants, 2021, doi:10.3390/plants10071321_

Round 1

Reviewer 1 Report

About the manuscript.... One of the problems that the authors should clarify concerns the identification of the pathogen: it was made on the basis of the symptoms of a simptomatic sample. This is not in the least correct. They should proceed according to coded identification methods and a good basis for identifying a fungus belonging to the genus Alternaria can be found published in the work of Woundenberg et al. 2013 (Studies in Micology 75: 171-212). It may seem like a small thing but it is essential if someone carries out experiments on wild isolates and not obtained from collections of microorganisms. If the identification is not done correctly the whole work may not have a solid basis and the results may not be reliable or comparable

Furthermore, the strain should be deposited in a collection and have a unique identification code ...

Short attention was also paid to Latin names, frequently not in italics .... 

Author Response

Dear reviewer,

Thank you very much for the time taken reviewing the manuscript.

In the present reviewed version of the manuscript (R1) a reference about how the pathogen isolation and identification was done has been included. This part was made by Dr. Alfredo Lacasa’s group. They are experts in this filed and they collected the samples, isolated and identified the fungus and kept it in a collection. They are experts on doing this part, we are not pathologists this is why we borrowed the strain from them.

This paper is not about pathology, is about how to select plant with plant breeding purposes by tissue culture. Field where we are experts in. We believe that after these clarifications the reviewer will reconsider his/her opinion about the reliability of our experiment.

On the other hand, the issue with the latin names is also solved in the revised version.

Reviewer 2 Report

The paper “Comparison of Four Systems to Test the Tolerance of ‘Fortune’ Mandarin Tissue Cultured Plants to Alternaria alternata” by Margarita Pérez-Jiménez and Olaya Pérez-Tornero is interesting and this is an important topic to better understand host and the pathogen Alternaria alternata relations. The authors compare several methods to test the tolerance of ‘Fortune’ mandarin tissue cultured plants to Alternaria alternata.

The paper is well written. It shows clear and some important results, although preliminary ones. It is well illustrated; the methods are clearly explained and the references are adequate.

The statistics is adequate.

It deserves publication in my opinion, even though the results being preliminary ones. I would consider this paper only as a short communication, not a full paper. Plants also accept Short Communications.

Some things:

Line 12. “Alternaria” in the abstract is not in italic. Please correct.

Also correct these same mistakes along the text (lines 28, 31, 32 and 33, 37 and so on…). Also, it happens again in the Results, Discussion, Materials and Methods and Conclusions sections…

How were the isolates of Alternaria alternata pv. citri identified? This should be mentioned in the materials and methods section (4.2 subsection). I think the authors got the isolate form someone, but the origin of the isolate must be referred and not only acknowledged in the “Aknowledgments”.

In Discussion, please rephrase lines 127-129. When you are referring to authors of reference 13, please write their names: “Only Chakraborty et al. (2020)…… (13).

See line 133. So, why did you not try also higher concentrations?

Lines 143 to 152. Why did you not analyse the filtrate? This could be important as you state. And if you consider this is not a good method, why did you try it? Maybe, using spores’ suspensions would be a better method by itself.

Please, rewrite lines 153-154 and explain better this statement.

The conclusions are clear and they resume the best findings of this preliminary study.

Author Response

Dear reviewer 1,

Thank you very much for your time reviewing the manuscript and your valuable comments.

English language and style: Moderate English changes required

The paper has been reviewer by a translator and reviewer for their improvement of the English language and style.

I would consider this paper only as a short communication, not a full paper. Plants also accept Short Communications.

We do not have any problem if this paper is published as a short communication instead of a research paper.

Line 12. “Alternaria” in the abstract is not in italic. Please correct. Also correct these same mistakes along the text (lines 28, 31, 32 and 33, 37 and so on…). Also, it happens again in the Results, Discussion, Materials and Methods and Conclusions sections…

“Alternaria” in the manuscript has been corrected and it is now in italics.

How were the isolates of Alternaria alternata pv. citri identified? This should be mentioned in the materials and methods section (4.2 subsection). I think the authors got the isolate form someone, but the origin of the isolate must be referred and not only acknowledged in the “Aknowledgments”

The published paper where the isolation and identification of the isolates is explained has been added. Nemsa, I.; Hernández, M.A.; Lacasa, A.; Porras, I.; García-Lidón, A.; Cifuentes, D.; Bouzid, S.; Ortuño, A.; Del Río, J.A. Pathogenicity of Alternaria alternata on fruits and leaves of ‘Fortune’ mandarin (Citrus clementina×Citrus tangerina). Can. J. Plant Pathol. 2012, 34:2, 195-202

In Discussion, please rephrase lines 127-129. When you are referring to authors of reference 13, please write their names: “Only Chakraborty et al. (2020)…… (13).

The lines 127-129 have been rephrased and Chakraborty et al. (2020) has been added to the discussion.

See line 133. So, why did you not try also higher concentrations?

We did not try higher concentrations because the ACT obtained was not sufficient to increase the concentration in the media. Tissue culture experiments require several repetitions and the culture media needed such an important quantity of ACT that was not possible to produce it even when the lab that produced was expert in this field.

Lines 143 to 152. Why did you not analyse the filtrate? This could be important as you state. And if you consider this is not a good method, why did you try it? Maybe, using spores’ suspensions would be a better method by itself.

We are planning to analyze the filtrate but there are many compounds than can be found. Maybe plant growth regulators, secondary metabolites and many other, so the analysis can be extended and as long as we want. On the other hand, we tried the method because is used in many other fungus diseases and we thought we should tried. Afterwards, we thought that we did not know the content of the filtrate and probably the results was not accurate enough to select plants that was our aim.

When a substance is added to the culture medium, in in vitro culture, it must be sterilized by filtration or in the autoclave, then spores would die. If a spores’ suspension without sterilize was added to the culture medium, very rich in sugar, the fungus would growth greatly and prevent the explant growing.

Please, rewrite lines 153-154 and explain better this statement.

Lines 153-154 have been rewritten

Reviewer 3 Report

The manuscript «Comparison of the four systems to verify the stability of cultivated plants mandarin« Fortune »to Alternaria Alternata» devoted to the important problem for agriculture.

The authors carried out an experimental study and developed a protocol for the selection of mandarin genotypes resistant to A. alternata. The experiment plan is well thought out and structured. The results are detailed and well illustrated. The conclusions are reasoned and beyond doubt.

Nevertheless, despite its great practical value, the scientific aspect of this study is clearly not finalized. The manuscript is too laconic and in its current form looks more like a report than a scientific article.

To improve the manuscript, the authors could expand the discussion section and perform a more detailed analysis of the literature, describing the pathogenesis mechanisms of A. alternata.

The methods are described quite well, but there are some inaccuracies.

Has this strain of A. alternata been sequenced? Is it deposited in the culture collection? A separate subsection could be devoted to the description of identification and storage of fungal cultures.

In my opinion, it would be appropriate to describe in more detail the ACT purification method, and not be limited to a reference.

I think this is an accidental mistake, but it is important to pay attention and correct in the title Alternaria Alternata to Alternaria alternata

Author Response

Dear reviewer,

Thank you very much for your time reviewing our manuscript. We really appreciate it. We have addressed the requirements and reply the queries.

Reviewer requests

Nevertheless, despite its great practical value, the scientific aspect of this study is clearly not finalized. The manuscript is too laconic and in its current form looks more like a report than a scientific article.

Reviewer 1 also has asked for changing classifications. Short communication instead of research article. We do not have any problem if editor agree with reviewers.

To improve the manuscript, the authors could expand the discussion section and perform a more detailed analysis of the literature, describing the pathogenesis mechanisms of A. alternata.

The pathogenesis mechanisms of A. alternata are described in many other papers about pathogenicity. In this paper we have described the comparison of 4 methods to select plant by tissue culture. The focus of this article is different and probably more technical as the reviewer says, this is why would prefer to change the article type as a short communication as both reviewers suggests. Changing the focus of the article would change also the meaning and the importance of our study.

The methods are described quite well, but there are some inaccuracies. Has this strain of A. alternata been sequenced? Is it deposited in the culture collection? A separate subsection could be devoted to the description of identification and storage of fungal cultures.

The details about the isolation, the culture collection and other aspects are described in a new reference that has been added. As we indicate in the manuscript, we did not isolate and identify the strain. The group of Dr. A. Lacasa that is expert in this field did it. This why we have added a new reference where everything is well explained. This paper is about plant selection by tissue culture, the pathogenic details are out of this paper scope and they must be explained by experts.

In my opinion, it would be appropriate to describe in more detail the ACT purification method, and not be limited to a reference.

We are sorry but we did not purified the ACT, the people from the article we quoted did it. This is why we reference to their paper.

I think this is an accidental mistake, but it is important to pay attention and correct in the title Alternaria Alternata to Alternaria alternata

We are sorry but we do not see Alternata Alternata in any title of our manuscript.

Reviewer 4 Report

The Manuscript, currently entitled "Comparison of Four Systems to Test the Tolerance of ‘Fortune’ Mandarin Tissue Cultured Plants to Alternaria Alternata", concerns the selection process under in vitro culture of Citrus reticulata cultivar to obtain lines with increased tolerance to Alternaria alternata pv. citri. The authors set themselves an important research goal and planned the experiments properly. However, the present manuscript still has some minor errors. Below in the points they are listed in accordance with the attached text

  1. The first one is already found in the title: we write the genre epithet in lower case, which the authors certainly know.
  2. Results, Figure 2: Illegible lettering on the photograph. Please, do better contrast the letters with the bacground
  3. Results, line 124: in the place of phrase 'to the shot leaves' insert: "leaves excised from shoot culture"
  4. Results, line144: Please, arrange appropriately the order of citations
  5. Results, line149-150: in the place of 'the exposition to the products of fungus growth as a filtrate included in the culture media' please insert: "the exposition to the products of the fungus activity in the filtrate applicated via culture media"
  6. Results, line 151: '...culture, fungus...'Please, break it down to two separate sentences at this point, i.e.: cultures. Fungus
  7. M&M, lines 170-176: Please, correct the notation of units
  8. M&M, line186,187: inadequate writing record of quoting literature
  9. References, 312: Incorrect form of citation

Author Response

Dear reviewer,

Thank you very much for your time reviewing our manuscript. We really appreciate it. We have addressed the requirements and reply the queries.

The first one is already found in the title: we write the genre epithet in lower case, which the authors certainly know.

                In the original manuscript the Alternaria alternata name is correctly written, but we have seen this mistake in the pdf that the reviewers have received. We can’t correct this, this is an editor thing.

Results, Figure 2: Illegible lettering on the photograph. Please, do better contrast the letters with the background

                If you maximise the figure there is not problem of legibility, maybe is also a matter of the pdf building.

Results, line 124: in the place of phrase 'to the shot leaves' insert: "leaves excised from shoot culture"

                This line has been corrected and therefore the section 4.5.

Results, line144: Please, arrange appropriately the order of citations

                This has been fixed.

Results, line149-150: in the place of 'the exposition to the products of fungus growth as a filtrate included in the culture media' please insert: "the exposition to the products of the fungus activity in the filtrate applicated via culture media"

It has been changed

Results, line 151: '...culture, fungus...'Please, break it down to two separate sentences at this point, i.e.: cultures. Fungus

                It has been changed

M&M, lines 170-176: Please, correct the notation of units

                It has been corrected

M&M, line186,187: inadequate writing record of quoting literature

                It has been corrected

References, 312: Incorrect form of citation

                It has been corrected.

Round 2

Reviewer 2 Report

I thank the authors for the corrections and improvments. All my major doubts and concerns were adressed. Nevertheless, I do keep my opinion that this paper is a Short Communication.

Minor things:

"Alternaria Alternata" in the title must be corrected to "Alternaria alternata".

Line 31. "Citrus reticulata" is not in italic, nor "Citrus" in line 37 nad they must be in italic. Please put all the latin names of the species or genera in italic. Check this carefully along the text. 

Also "Alternaria citri" in the legends of the figures...(lines 102, 106 and 110).

Lines 180 and 181. "...as described by..." Please refer the authors' names and then put the number of the reference (23) and (24), as you did in line 129. Check this please!

Author Response

Dear reviewer,

Thank you very much for your time reviewing our manuscript. We really appreciate it. We have addressed the requirements and reply the queries.

"Alternaria Alternata" in the title must be corrected to "Alternaria alternata".

                In the original manuscript the Alternaria alternata name is correctly written, but we have seen this mistake in the pdf that the reviewers have received. We can’t correct this, this is an editor thing.

Line 31. "Citrus reticulata" is not in italic, nor "Citrus" in line 37 nad they must be in italic. Please put all the latin names of the species or genera in italic. Check this carefully along the text.

                We have changed and now all the “latin” names are in italic. However, when the name of a group of plants (family, genus…) get colloquial, then it is not compulsory to write it in italics.

Also "Alternaria citri" in the legends of the figures...(lines 102, 106 and 110).

                It has been corrected.

Lines 180 and 181. "...as described by..." Please refer the authors' names and then put the number of the reference (23) and (24), as you did in line 129. Check this please!

                This has been also corrected.
